# An inhibitor-free, versatile, fast, and cheap precipitation-based DNA purification method

**Zhe F. Tang, David R. McMillen***

Department of Chemical and Physical Sciences, University of Toronto, Mississauga, Ontario, Canada

* david.mcmillen@utoronto.ca

## Abstract

Nucleic acid purification is a key step in molecular biology workflows, and especially critical in synthetic biology. Two common techniques are phenol chloroform extraction and silica column adsorption. We have determined that commercial silica columns appear to elute a currently unidentified substance that can inhibit subsequent enzymatic reactions if not sufficiently diluted. To resolve this inhibition, we have developed a novel purification approach in which we achieve simultaneous protein removal and DNA precipitation through the application of chaotropic salts and alcohol/polyethylene glycol. While prior DNA precipitation approaches require 2 steps to remove protein and precipitate DNA, and 4 steps to remove RNA and precipitate DNA, our method accomplishes all of them in a single step. Our approach matches the speed and versatility of silica column purification while additionally being substantially cheaper, as well as avoiding restrictions on the maximum size of purified DNA fragments and the need for gel extraction to remove primer dimers below 700 bps. Our purification technique has also enabled us to uncover an important insight into nucleic acids: Gibson Assembly generates mainly linear DNA that transforms poorly into the cloning host E. coli, which is linked to suboptimal levels of functional colony formation after transformation. We show that decreasing the concentration of the linear DNA by 100-fold dramatically increases circularization.

**Data availability statement:** All files containing the raw experimental results are available from the Harvard Dataverse repository, freely accessible at https://doi.org/10.7910/DVN/GYNVHE. The data is organized by Figure,

## Introduction

Nucleic acid purification is a key step in molecular biology, and is particularly prominent in synthetic biology workflows [1]. Projects in synthetic biology often involve goals such as the production of useful chemical molecules [1] or the construction of novel regulatory networks with designed and tuned properties [2]. Attaining these goals generally involves the testing of many recombinant DNA "prototypes" in which multiple sequence candidates are assembled and tested. These candidates may be different metabolic pathway enzymes or different components of regulatory pathways. A typical workflow for the creation of a recombinant DNA prototype [1] involves:

1. amplification of DNA containing the desired parts using PCR or culturing *E. coli* that contains the parts;

and is best viewed in "Tree" mode, which will display a branch for each Figure, providing files containing the data used in each panel in the Figure.

**Funding:** Natural Sciences and Engineering Research Council (NSERC) of Canada, grant number RGPIN-2017-06795.

**Competing interests:** The authors have declared that no competing interests exist.

2. purification of the DNA parts to prevent enzymatic inhibition in subsequent steps;
3. enzymatic assembly of the DNA parts into a plasmid;
4. transformation of the assembly reaction into a microbial host;
5. purification of plasmid DNA from microbial colonies where each colony contains a possibly differentially assembled plasmid;
6. sequencing of the purified DNA to identify a correctly-assembled DNA prototype;
7. functional characterization of the DNA prototype to see if the desired properties have been achieved.

This workflow requires at least two DNA purification steps.

After a possible cell lysis step to release DNA and proteins from the cell membrane (Fig 1A), two commonly used techniques to purify DNA are phenol chloroform extraction (Fig 1C) and silica column adsorption of nucleic acids (Fig 1D) [3]. Phenol chloroform extraction first creates an emulsion of phenol-chloroform with the sample and then separates proteins from the DNA into the organic phase via a centrifugation step. The nucleic acid in the protein depleted aqueous fraction is finally precipitated in the presence of a salt and either an alcohol or polyethylene glycol. On the other hand, silica columns selectively adsorb nucleic acids in the presence of chaotropic salts, allowing protein to pass through the silica column to be discarded. The DNA bound to the silica column is then eluted from the column in water or T10E0.1 buffer (10 mM Tris-HCl, 0.1 mM EDTA, pH8).

Phenol chloroform extraction produces high quality nucleic acid but is known to be labor intensive, which was a significant factor in the widespread adoption of silica columns. Common descriptions of phenol chloroform extraction are not explicit about which aspects of the protocol are most time-consuming. There are two stages in the procedure that take similar amounts of time: organic extraction and nucleic acid precipitation. In the first stage, two organic extraction steps are performed to minimize the presence of highly denaturing phenol in the aqueous phase. The first extraction uses 1:1 phenol and chloroform as the organic solvent. Although a bottom organic layer and top aqueous layer is formed after centrifugation, the top aqueous layer actually retains a non-negligible amount of phenol because the solubility of phenol in water is 8.4g per 100mL [5]. Because the solubility of chloroform in water is only 0.8g per 100 mL [6], a second organic extraction using only chloroform removes most of residual phenol in the aqueous phase. In each organic extraction step, care (and thus time) is typically taken to slowly pipette out the aqueous phase to minimize carry over of the organic phase. Additionally for this stage, the time it takes to perform 2+2 transfers of the samples to-and-from the centrifuge and labeling 2+1 fresh tubes is not negligible.

In this article, we report a vastly simplified DNA precipitation approach that not only addresses previously unreported pitfalls of silica column purification, but also matches its speed and versatility. Specifically, we have discovered that commercial silica columns elute a low level of an unidentified substance that inhibits subsequent enzymatic reactions if not sufficiently diluted. To resolve this enzyme inhibition, we have discovered that the presence of chaotropic salts and alcohol / polyethylene glycol (PEG) enables simultaneous protein removal and nucleic acid precipitation, resulting in a novel nucleic purification method. Our approach has four steps (Fig 1B): simultaneous protein deactivation and nucleic acid precipitation; an extra protein dilution step to remove any remaining active protein; removal of the chaotropic salts; and resuspension of the nucleic acid pellet. Simultaneous protein removal and DNA precipitation is much simpler than the multiple steps required in previously published DNA purification methods [4,7] (Fig 1E). Compared to silica column purification, our purification method not only successfully resolves silica column based enzyme inhibition, but

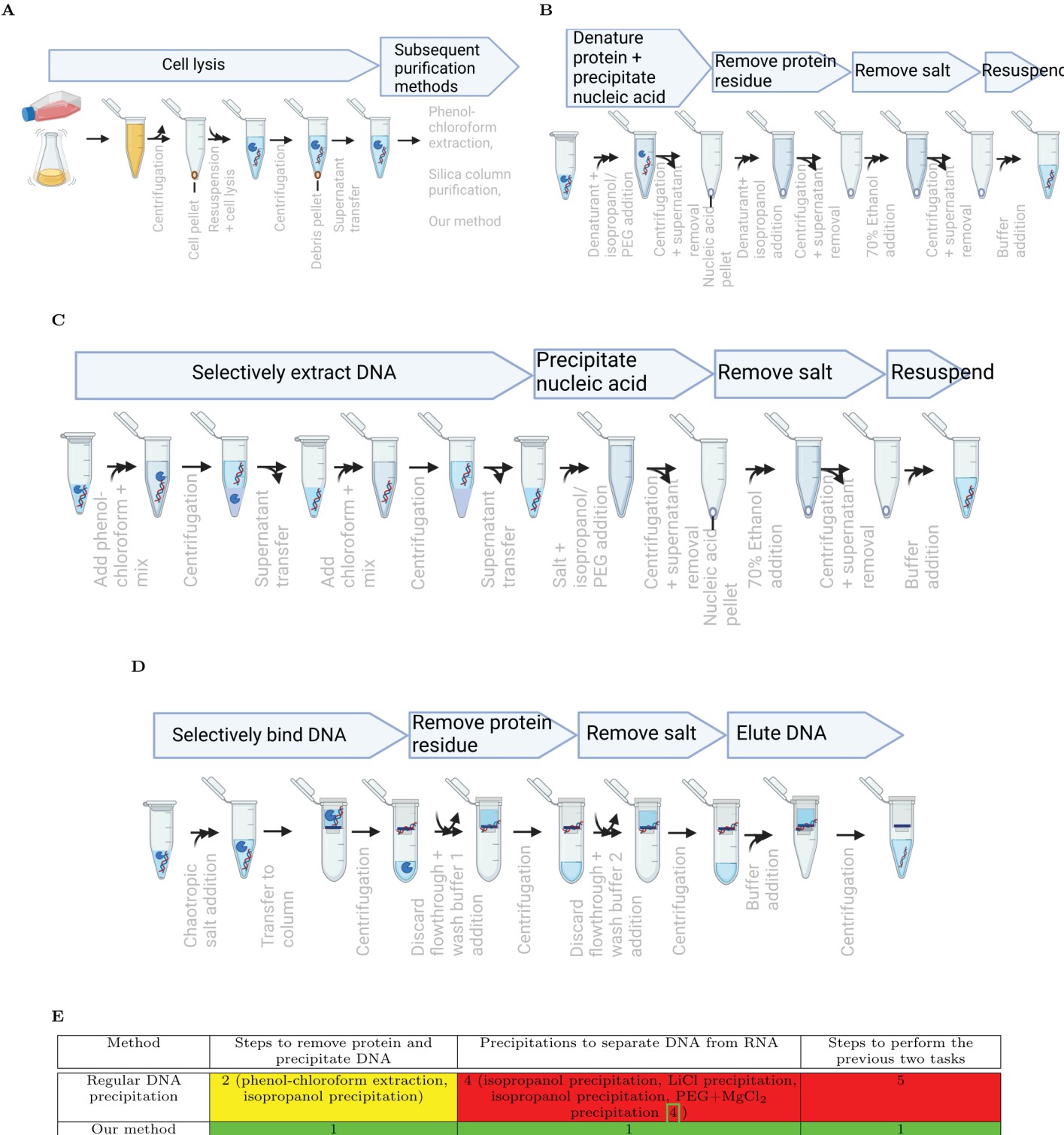

**Fig 1. Overview of different DNA extraction methods.** Illustrations were created with BioRender.com. (A) Cell lysis to release DNA from the cells. (B) Our purification method. (C) Phenol-chloroform extraction. (D) Silica column purification. (E) Simplicity of our purification method compared to previously published DNA purification methods.

does not have upper size restrictions, can avoid using gel extraction for primer dimers below 700 bps, and is substantially cheaper.

## Results and discussion

### Our novel purification procedure simultaneously inactivates proteins and precipitates DNA

We discovered that silica column purified single-copy plasmid DNA resulted in poor sequencing results if the eluent was not sufficiently diluted. Purification of single-copy plasmid pBeloBac11 using Qiagen's recommended procedure of lysing 5 mL of culture per column only resulted in a sufficient amount of DNA for one sequencing reaction. After using vacuum centrifugation to dry down the water-eluted sample to make it sufficiently concentrated for Sanger sequencing, we observed very low numbers of reads (Fig 2A: QM.5). Increasing the amount of lysed cell per column to 10 mL according to Qiagen's supplementary protocol for low copy plasmids (Fig 2A: QM.10) and Re-Column purification of DNA pooled from 5 columns to avoid drying down the sample (Fig 2A: QM.RC) still resulted in low numbers of reads. We obtained close to the expected number of DNA sequencing reads when lysate from 50 mL of culture was loaded onto the same column such that the fraction of eluent in the sequencing sample was sufficiently low (Fig 2A: QM.OLH and QM.OLL). QM.OLL used 1/2 the amount of the DNA sample per sequencing reaction compared to QM.OLH, so the amount of sequencing inhibition is expected to be lower.

One downside of the column overloading strategy is the increased opportunity for genomic DNA to outcompete plasmid DNA for the finite amount of silica surface, under suboptimal cell lysis conditions (Fig 2B). In such cases, the presence of plasmid bands (indicated by the arrows in Fig 2B) is very faint compared to the high molecular weight genomic DNA bands. Since the column overloading strategy may not always be advantageous, we wondered if there was an alternative purification strategy to improve the quality of DNA extractions without resorting to Sambrook's [4] laborious phenol-chloroform purification.

While existing literature separately described the ability of high concentrations of chaotropic salt to denature proteins and decreased nucleic acid solubility in salt-containing alcohol or PEG, we discovered that chaotropic salts combined with isopropanol or PEG could denature proteins and selectively precipitate nucleic acids in one convenient step. Prior DNA precipitation methods require a distinct step for each of protein removal and DNA precipitation (Fig 1E). Prior to our discovery, one concern is that the concentration of a saturating solution of chaotropic salt is decreased upon addition of precipitating agents such as isopropanol and PEG, likely allowing the protein to regain some activity. Additionally, sufficiently high concentrations of a strong chaotropic salt in the presence of isopropanol resulted in large loss of DNA after precipitation (Fig 2C: GITC 2M + 50% Isop) compared to a lower concentration of the strong chaotropic salt (Fig 2C: GITC 1M + 50% Isop) or a high concentration of a weaker chaotropic salt (Fig 2C: GHCl 2.7M + 50% Isop), consistent with reports of chaotropic salts denaturing double stranded DNA as well as proteins [9]. Decreased centrifugation temperature resulted in increased DNA recovery (Fig 2C, GITC 2M + 50% Isop + Gentle Mixing + 4C centrifugation), consistent with decreased temperature facilitating renaturation of DNA [10]. After recognizing those novel findings, we identified suitable concentrations of both chaotropic salts that eliminate proteinase K activity to undetectable levels for both precipitation agents isopropanol and PEG, with protein dilution solution further eliminating proteinase K activity left over from DNA precipitation (Fig 2E: top part). Residual proteinase K activity is measured by the sample's inhibition of Ampligase-mediated ligation of

**A**

| Column Name | Ctrl | QM.5 | QM.5 | QM.10 | QM.10 | Ctrl | QM.5 | QM.RC | QM.RC | QM.OLH | QM.OLH | QM.OLL | QM.OLL |
|---|---|---|---|---|---|---|---|---|---|---|---|---|---|
| Purification method | Ctrl | QM | QM | QM | QM | Ctrl | QM | QM | QM | QM | QM | QM | QM |
| Day | 1 | 1 | 1 | 1 | 1 | 2 | 2 | 2 | 2 | 2 | 2 | 2 | 2 |
| Volume of cells per column (mL) | NA | 5 | 5 | 10 | 10 | NA | 5 | 5×10 | 5×10 | 50 | 50 | 50 | 50 |
| Fraction of eluent in sequencing reaction | NA | 1 | 1 | 0.5 | 0.5 | NA | 1 | 0.1 | 0.1 | 0.1 | 0.1 | 0.05 | 0.05 |
| DNA concentration (ng / 100 bp / sequencing reaction) | 1.08 | 3.84 | 3.02 | 1.53 | 1.90 | 1.39 | 1.58 | 1.39 | 1.64 | 2.88 | 2.82 | 1.44 | 1.41 |
| Read length (bp) | 1000 | 157 | 0 | 321 | 270 | 1000 | 0 | 60 | 170 | 750 | 595 | 750 | 750 |
| Compact letter display [8] (see Methods) | a | b | | | | | b | | | c | | ac | |

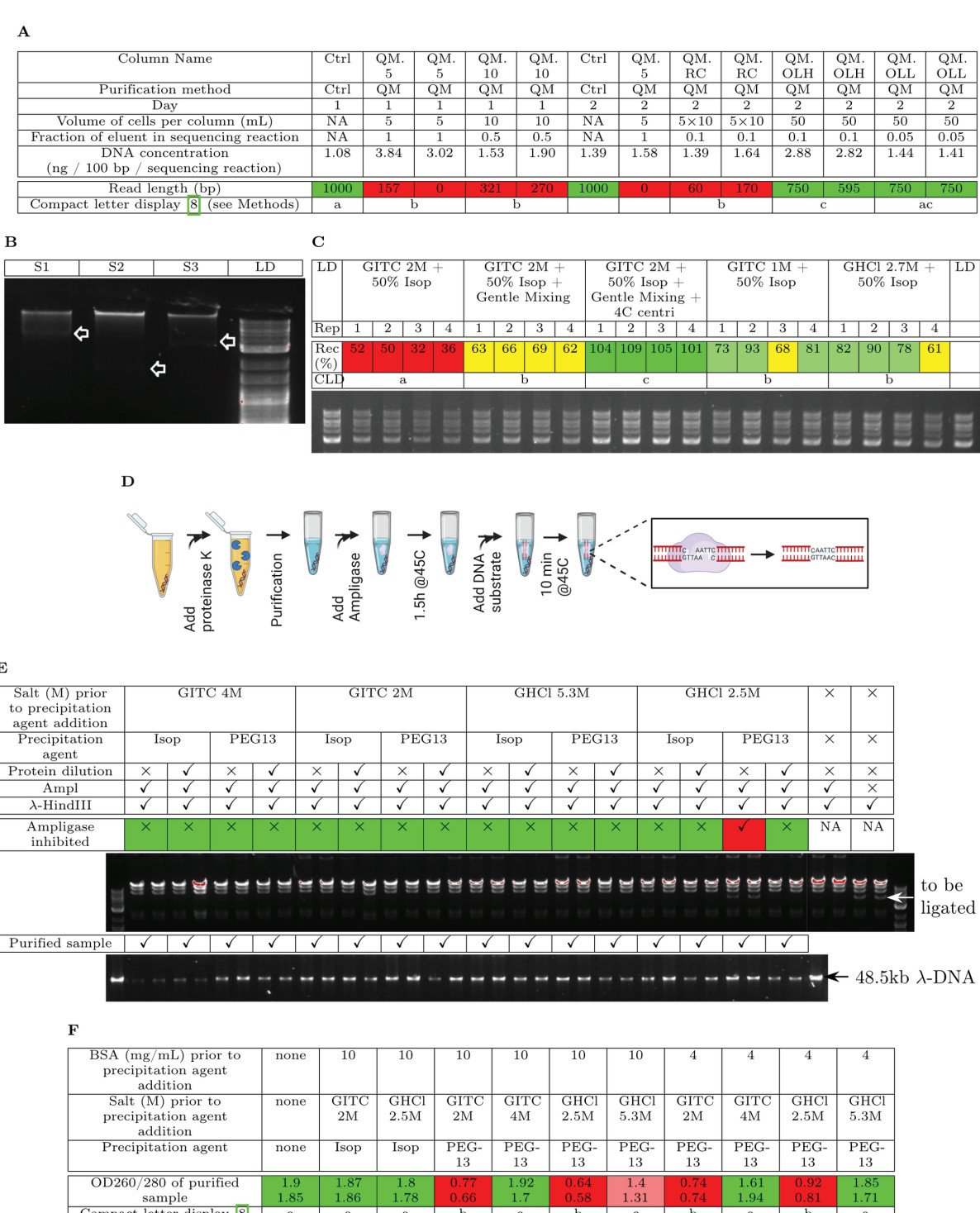

**B**

S1 | S2 | S3 | LD

**C**

| LD | GITC 2M + 50% Isop | | | | GITC 2M + 50% Isop + Gentle Mixing | | | | GITC 2M + 50% Isop + Gentle Mixing + 4C centri | | | | GITC 1M + 50% Isop | | | | GHCl 2.7M + 50% Isop | | | | LD |
|---|---|---|---|---|---|---|---|---|---|---|---|---|---|---|---|---|---|---|---|---|---|
| Rep | 1 | 2 | 3 | 4 | 1 | 2 | 3 | 4 | 1 | 2 | 3 | 4 | 1 | 2 | 3 | 4 | 1 | 2 | 3 | 4 | |
| Rec (%) | 52 | 50 | 32 | 36 | 63 | 66 | 69 | 62 | 104 | 109 | 105 | 101 | 73 | 93 | 68 | 81 | 82 | 90 | 78 | 61 | |
| CLD | a | | | | b | | | | c | | | | b | | | | b | | | | |

**D**

Add proteinase K → Purification → Add Ampligase → 1.5h @45C → Add DNA substrate → 10 min @45C

**E**

| Salt (M) prior to precipitation agent addition | GITC 4M | | | | GITC 2M | | | | GHCl 5.3M | | | | GHCl 2.5M | | | | × | × |
|---|---|---|---|---|---|---|---|---|---|---|---|---|---|---|---|---|---|---|
| Precipitation agent | Isop | | PEG13 | | Isop | | PEG13 | | Isop | | PEG13 | | Isop | | PEG13 | | × | × |
| Protein dilution | × | ✓ | × | ✓ | × | ✓ | × | ✓ | × | ✓ | × | ✓ | × | ✓ | × | ✓ | × | × |
| Ampl | ✓ | ✓ | ✓ | ✓ | ✓ | ✓ | ✓ | ✓ | ✓ | ✓ | ✓ | ✓ | ✓ | ✓ | ✓ | ✓ | ✓ | × |
| λ-HindIII | ✓ | ✓ | ✓ | ✓ | ✓ | ✓ | ✓ | ✓ | ✓ | ✓ | ✓ | ✓ | ✓ | ✓ | ✓ | ✓ | ✓ | ✓ |
| Ampligase inhibited | × | × | × | × | × | × | × | × | × | × | × | × | × | × | ✓ | × | NA | NA |
| Purified sample | ✓ | ✓ | ✓ | ✓ | ✓ | ✓ | ✓ | ✓ | ✓ | ✓ | ✓ | ✓ | ✓ | ✓ | ✓ | ✓ | | |

to be ligated

← 48.5kb λ-DNA

**F**

| BSA (mg/mL) prior to precipitation agent addition | none | 10 | 10 | 10 | 10 | 10 | 10 | 4 | 4 | 4 | 4 |
|---|---|---|---|---|---|---|---|---|---|---|---|
| Salt (M) prior to precipitation agent addition | none | GITC 2M | GHCl 2.5M | GITC 2M | GITC 4M | GHCl 2.5M | GHCl 5.3M | GITC 2M | GITC 4M | GHCl 2.5M | GHCl 5.3M |
| Precipitation agent | none | Isop | Isop | PEG-13 | PEG-13 | PEG-13 | PEG-13 | PEG-13 | PEG-13 | PEG-13 | PEG-13 |
| OD260/280 of purified sample | 1.9 / 1.85 | 1.87 / 1.86 | 1.8 / 1.78 | 0.77 / 0.66 | 1.92 / 1.7 | 0.64 / 0.58 | 1.4 / 1.31 | 0.74 / 0.74 | 1.61 / 1.94 | 0.92 / 0.81 | 1.85 / 1.71 |
| Compact letter display [8] | a | a | a | b | a | b | c | b | a | b | a |

**Fig 2. Chaotropic-salt-based DNA precipitation recovers most DNA without protein contamination.** (A) Sequencing read length of plasmid samples purified using QIAprep Miniprep (QM) columns. Ctrl means a control plasmid that has previously resulted in high-quality sequencing data using the same primer. Please see S2A Fig for a visual comparison of sequencing read length. (B) Silica column extraction of plasmid DNA from three samples under suboptimal lysis conditions. (C) Recovery of 1kb plus ladder from NEB at different chaotropic salt concentrations, in the presence of 0.5 mg/mL of BSA. Following the precipitation, DNA pellets were only washed once in 70% ethanol + 5 mM MgCl$_2$. Please see S2B Fig for a visual comparison of percent DNA recovery. (D) Overview of the proteinase K deactivation assay. (E) Proteinase K deactivation of our purification method. When Ampligase is not inhibited by proteinase K presence, ligation of λ-HindIII DNA by Ampligase results in disappearance of the 4.36 kb band. (F) Assessing BSA contamination in samples purified using different chaotropic salts and different precipitation agents. Please see S2C Fig for a visual comparison of OD260/OD280 ratios.

the $\lambda$-HindIII ladder (Fig 2D), so no Ampligase inhibition is equivalent to no detectable proteinase K activity. Those purification methods successfully recovered enough $\lambda$-DNA to carry over proteinase K to inhibit subsequent enzymatic reactions (Fig 2E: bottom part). Lastly, sufficient concentrations of chaotropic salt in a mixture of BSA and DNA precipitated undetectable amount of BSA, with isopropanol leaving more BSA behind at higher maximum BSA concentration and lower chaotropic salt concentration compared to PEG (Fig 2F). The presence of unacceptable amount of BSA in the purified sample is determined by deviation in OD260/280 ratio from 1.7-2.0 [11].

## Our purification procedure offers advantages over silica column purification, while matching its speed and versatility

Our purification approach offers several advantages over silica column purification. First, consistent with the previously observed inhibition of DNA sequencing, sufficiently concentrated eluent from a silica column but not samples purified using our method resulted in inhibition of DNA sequencing (Fig 3B) and T5 exonuclease (Fig 3C: QQ/NEB vs GHCl + Isop/GHCl + PEG). Dilution of the eluent from a silica column relieved inhibition of T5 exonuclease (Fig 3C: NEB with 24ul eluent). In contrast to Jue et al.'s [12] attempt to alleviate enzymatic inhibition of silica column purified samples, inclusion of an 1-undecanol wash step prior to elution still inhibited T5 exonuclease when the eluent was sufficiently concentrated (Fig 3C: QQ + UD). The eluent concentration in our T5 digestion reactions was at least 20× higher than the eluent concentration in Jue et al.'s [12] qPCR reactions (6/7.35 vs $\frac{4}{50} \times \frac{4}{10}$). In addition, while Jue et al [12] observed inhibition of qPCR reactions with Zymo Quick-DNA/RNA but not QIAquick, we observed silica column inhibition of enzymes for both NEB DNA Cleanup and QIAquick at much higher eluent concentrations. Since the samples purified using our approaches contained additional $\lambda$-genomic DNA that is digestable by T5 exonuclease, it is remarkable that the degree of DNA ladder digestion for samples purified using our method matched or exceeded the least inhibited samples purified using silica columns.

Second, compared to silica columns, our purification approach can selectively recover DNA above certain user-defined size thresholds and recover longer pieces of DNA. Silica columns typically retain >=100 bp fragments (Fig 3D), which is often smaller than the size of primer dimers. On the other hand, our purification method allowed us to customize the minimal size of DNA fragments that are efficiently recovered by altering the concentration of PEG 8000 and changing the choice of chaotropic salt (Fig 3E). Although size-selective PEG precipitation has been previously demonstrated in the presence of NaCl [13,14] and MgCl$_2$ [15], we provide novel characterization of how PEG selectively precipitates DNA in the presence of both chaotropic salts to avoid unnecessary salt exchange steps. In addition, QiaPrep silica columns resulted in smaller DNA sizes than our purification method when purifying a 10.7 kb plasmid (Fig 3F), while both methods resulted in identically sized DNA for purification of a 7.5 kb plasmid (Fig 3G). Since a small fraction of QiaPrep's purified product still aligns with the majority product from our purification method, QiaPrep's majority product is likely sheared linear DNA. The advantage of high molecular weight DNA precipitation can further be combined with size-thresholding ability of PEG precipitation, with PEG precipitated plasmid DNA leaving much more RNA behind than isopropanol precipitation (Fig 3G). Previously, removal of RNA from plasmid DNA in precipitation-based purification methods utilized extra precipitation steps such as LiCl precipitation followed by isopropanol precipitation and PEG-MgCl2 precipitation [4] (Fig 1E). Our purification method efficiently separates

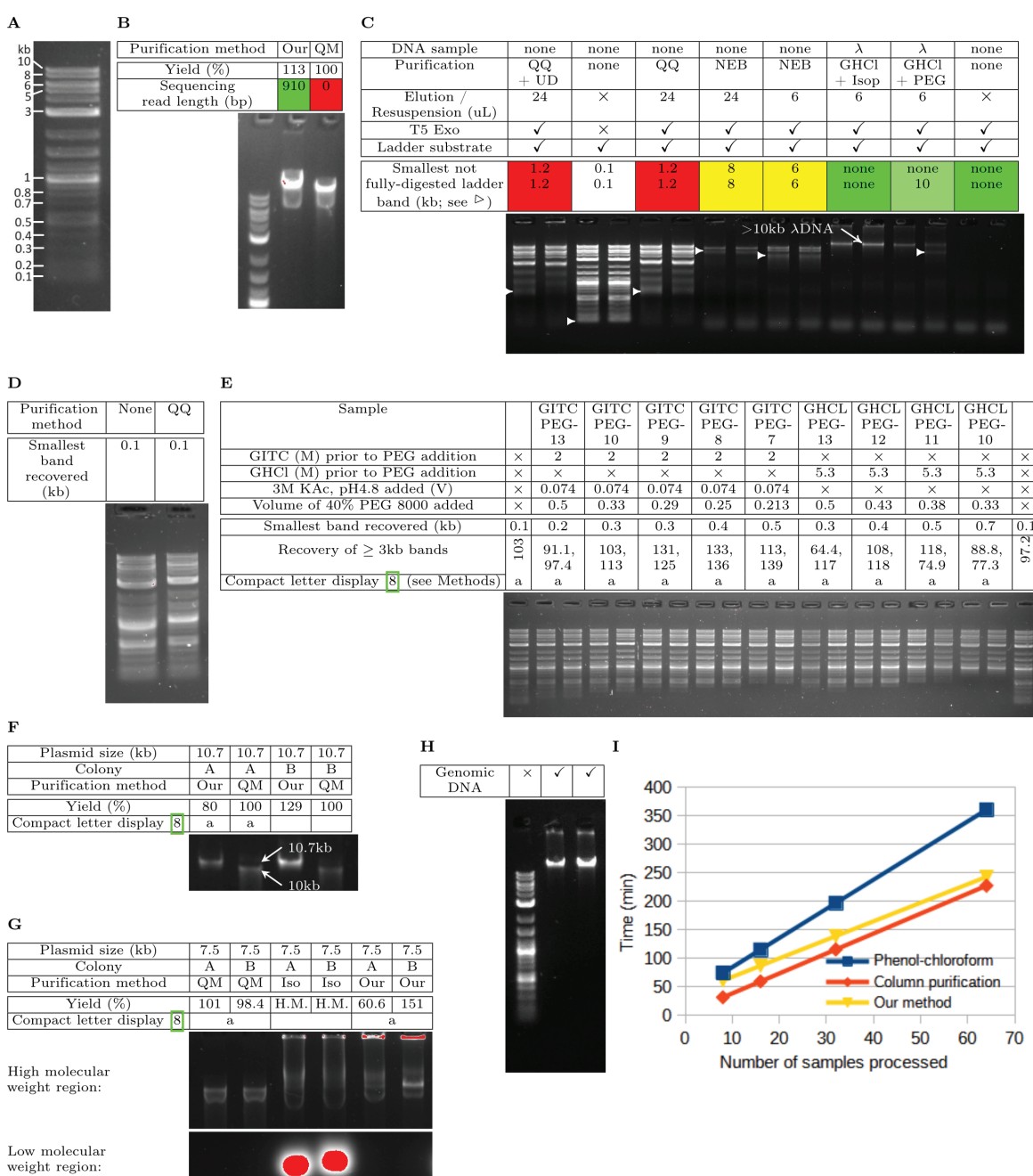

**Fig 3. Advantages of our purification method over silica column purification and phenol-chloroform extraction.** (A) Sizes of select bands in the 1kb+ ladder for quick reference. (B) Comparing Sanger sequencing read length for pBeloBac11 (New England Biolabs) between QiaPrep Miniprep (QM) and our purification method (Our). There is a large difference in sequencing read-length despite both methods having similar plasmid extraction yield. (C) Inhibition of T5 exonuclease by silica column purification (QQ, NEB) and our purification method (Our). UD means UnDecanol wash step between the final alcohol based wash step and 1 min centrifugation to remove residual wash solution from the column. (D) Purification of 1kb+ ladder using the QIAquick silica column. (E) Changing the minimum size of precipitated DNA by switching chaotropic salt and modulating PEG 8000 concentrations. Please see S2D Fig for a visual comparison of percent recovery of ≥ 3kb bands. (F) Purification of a 10.7 kb plasmid (Supplementary Data SB.gb) using different methods. Please see S2E Fig for a visual comparison of yield. (G) Purification of a 7.5 kb plasmid (pBeloBac11, New England Biolabs) using different methods. Iso means isopropanol precipitation rather than PEG precipitation following alkaline lysis of E coli cells and clarification of neutralized lysate. That is, 1 volume of isopropanol rather than 1/3 volume of PEG 8000 is mixed with the transferred neutralized supernatant. H.M. means hard to measure accurately, because the presence of high intensity low molecular weight blob often decreases the amount of nucleic acid intercalating dye available for high molecular weight bands in pre-stained agarose gels. Please see S2F Fig for a visual comparison of yield. (H) Genomic DNA extraction from *E. coli* MG1655 using our purification method. (I) Comparing protocol time between phenol-chloroform extraction, silica column purification, and our purification method.

higher molecular weight DNA and partially-digested lower molecular weight RNA in a single rather than multiple precipitation steps.

Lastly, compared to silica column purification, our purification method is just as versatile, is only 20% slower (Fig 3I), but is over 10 times cheaper (Table 1). Our purification method has successfully purified PCR products free from non-specific amplicons under 700 bp (Fig 3E), purified plasmid DNA mostly free of RNA (Fig 3G), and genomic DNA mostly free of RNA (Fig 3H). Our purification method becomes much faster than phenol-chloroform extraction and approaches the speed of silica column purification at higher number of samples (Fig 3I). When purifying 32 samples simultaneously, our purification method takes about 30% less time compared to phenol-chloroform extraction and takes only about 20% longer than silica column purification. To estimate protocol time to process an arbitrary number of samples, each task was timed for several samples and divided by that number if the task time is proportional to the number of samples (Supplementary data ProtocolTimeComparison-formanuscript.ods). For instance, loading samples into the centrifuge is roughly proportional to the number of samples, but the centrifugation time to precipitate DNA is constant. In addition, the cost of our purification method is less than 10% of the cost of using commercial silica column kits (Table 1).

## Our purification method enabled us to discover that most assembled plasmid-length DNA are in the linear form

Our purification technique has demonstrated advantages over existing DNA purification techniques (Fig 4). Compared to silica columns, our purification method resulted in a much less inhibitory sample at low eluent dilutions, did not have upper size restrictions, can avoid using gel extraction for primer dimers below 700 bps, and is over 10 times cheaper. Compared to phenol-chloroform extraction, our purification technique is noticeably faster when processing more than 25 samples simultaneously. Some of those advantages can be overcome with implementable workarounds. For instance, enzyme inhibition at low eluent dilution can be overcome with column overloading, at the increased risk of undesired DNA outcompeting desired DNA for silica binding; upper size restrictions on DNA length could be overcome by switching to the more time-consuming anion-exchange columns; primer dimers greater than 100 bp could be removed using the more time-consuming gel extraction; silica columns could be used instead of phenol-chloroform extraction to speed up the purification process, but introduces inhibitory substances that must be sufficiently diluted. However,

**Table 1. Comparison of cost per reaction in Canadian Dollars between our method and silica column purification according to UofTMedStore.**

| Individual items | Our purification method | Silica column purification |
|---|---|---|
| GHCl | $0.0816 ($40 per 100g, 267 μl of 8M GHCl used) | 0 |
| PEG 8000 | $0.0021 ($35 per 500g, 75 μl of 40% PEG 8000 used) | 0 |
| Isopropanol | $0.02 ($25 per 500 mL, 0.4 mL used) | 0 |
| MgCl2 | $0.000025 ($25 per 500g, 2.5 μl of 1 M used) | 0 |
| Anhydrous ethanol | $0.0119 ($17 per 500 mL, 0.35 mL used) | 0 |
| QIAquick | 0 | $2.12 ($530 per 250 reactions) |
| Total | $0.1156 | $2.12 |

UofTMedStore is a key supplier of molecular biology consumables to the University of Toronto's research community that offers discounted pricing for research. The cost of the smallest bulk chemical was used to calculate the per-reaction cost of our purification method. The largest and therefore most economical QIAquick purification kit was used to calculate the per-reaction cost of silica column purification.

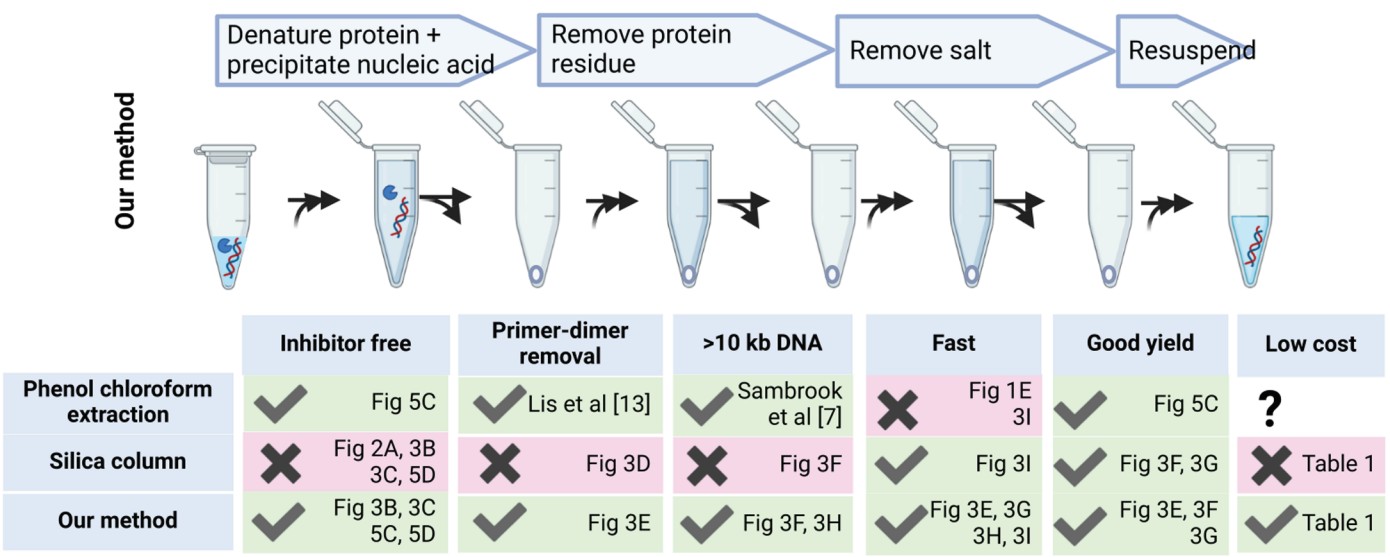

**Fig 4.  A summary of comparisons between different DNA purification methods.**

we show in this section that our purification method also allowed us to more easily discover important insights regarding DNA assembly, compared to both silica column purification and phenol-chloroform extraction.

Transformation efficiency of chemically competent cells prepared and transformed using Inoue's method [16] is usually $1 \times 10^8$ to $1 \times 10^9$ cfu per ug of pUC19. However, all reported colony outputs of *in vitro* assembly reaction mixtures are only in the thousands of colonies per 10 ng of reactant components at the most, rather than up to 10 million colonies. This mismatch in the colony output of purified plasmid DNA and assembly reaction mixture made us wonder whether the assembly efficiencies of current DNA assembly techniques is extremely low and how we could eliminate bottlenecks to such low assembly efficiency.

We assembled four DNA fragments using Gibson Assembly (NEB HiFi Assembly), analyzed some of the assembly reactions on an agarose gel, and transformed the remaining reaction mixture into chemically competent *E. coli*. The positions of detectable fragments on the gel matches the sizes of consecutive subsets of the initial four DNA fragments (Fig 5A). We measured the amount of DNA at the position matching the expected size of the complete plasmid (the product band) and computed the actual transformation efficiency of the product band in terms of cfu per μg of product band. Although prior literature on Gibson Assembly characterized size distribution of assembled products [17,18], none of them quantitatively mapped the amount of product-length DNA to colony forming units to see if the expected colony output is attained. The only criteria in prior characterizations is whether there is a sufficient number of colonies to increase probability of identifying a correctly assembled clone (Fig 5A). Increasing the number of picked colonies isn't always the best option, as the number of picked colonies must be balanced against the cost of sequencing for all picked colonies. In our study, we observed a greater than 100-fold reduction in actual transformation efficiency of the product band compared to predicted transformation efficiency of the product band if the product band consists entirely of the same circular plasmid isolated from *E. coli* (Fig 5A).

We suspected that most of the product band we saw on the gel were not circular DNA and the assembly reaction resulted in very little circular DNA. To distinguish between completely

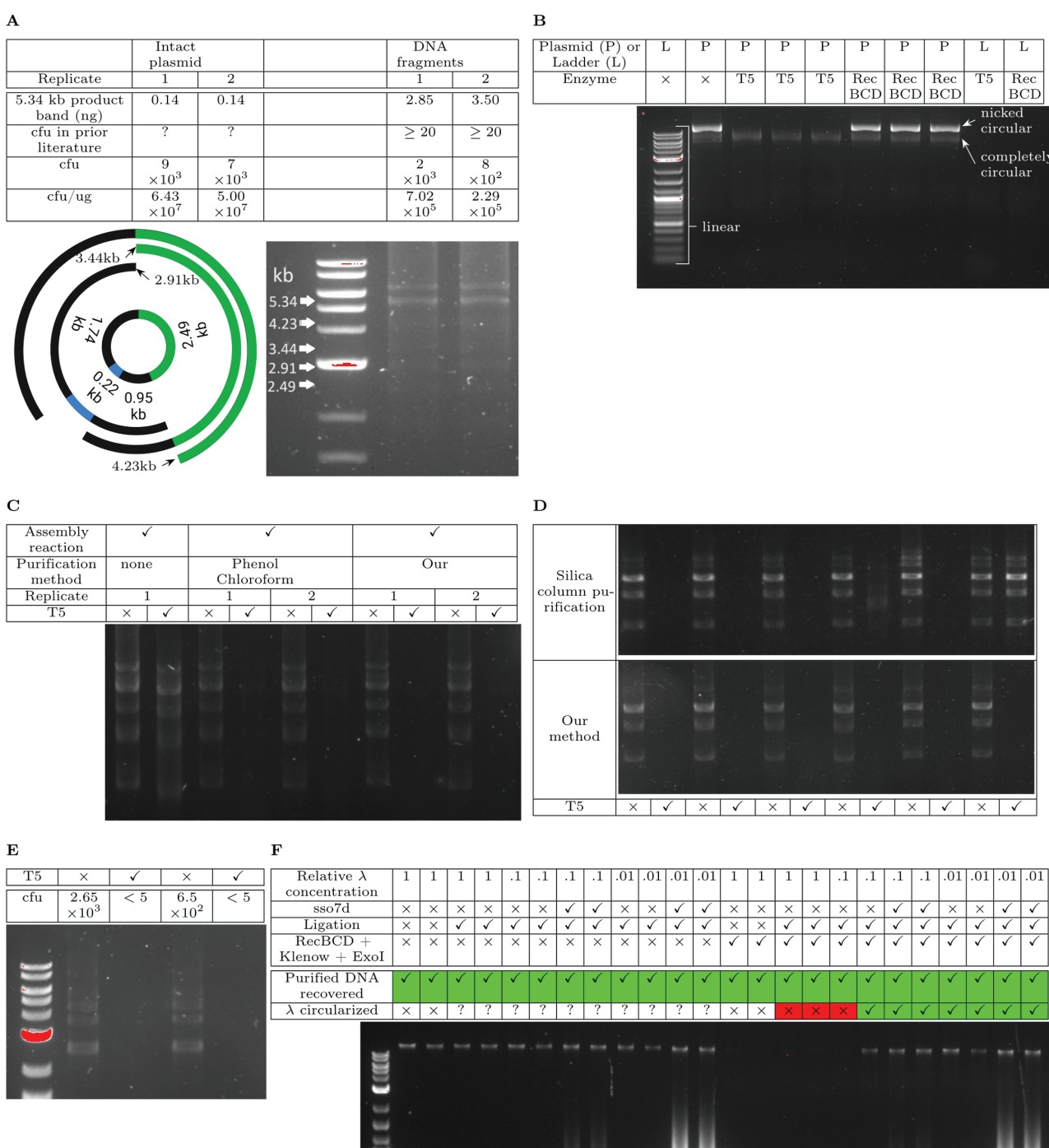

**Fig 5. New insight: most assembled plasmid DNA exists in linear form.** (A) Expected and observed transformation efficiency of Gibson Assembly of a 5.34 kb plasmid (Supplementary Data B7M0.gb). (B) Activity of T5 exonuclease and RecBCD on a purified plasmid (Supplementary Data C8QEMBS.gb) and linear DNA ladder. (C) T5 exonuclease digestion after purification of Gibson Assembly of plasmid B7M0 using different methods. (D) Comparison of T5 exonuclease activity in assembly reactions that are purified using silica columns and our method. (E) Comparison of colony output for purified Gibson Assembly reactions of plasmid B7M0 in presence and absence of T5 exonuclease. (F) Effect of different $\lambda$-DNA concentrations and sso7d on circularization of $\lambda$-DNA. "?" means not yet determined. Identical total amount of $\lambda$-DNA is present in each reaction.

circular and not-completely circular (linear or nicked circular) DNA, we purified the assembly mixture using our technique and digested the purified assembly mixture using T5 exonuclease. While T5 exonuclease completely digested the linear DNA ladder and nicked circular plasmids, it left completely circular plasmid DNA intact ( Fig 5B), consistent with prior literature [19,20]. We also verified the necessity of DNA purification for T5 exonuclease to completely digest linear DNA. Without purification of the assembled reaction, a large amount of DNA was left over after T5 digestion (Fig 5C). Column purification of the assembled reactions also resulted in large variability in leftover DNA between replicate reactions (Fig 5D). On the other hand, T5 digestion of assembly reactions, that were purified using our method, resulted in no detectable fully circular DNA (Fig 5C, 5D, and 5E) and more than 100-fold decrease in transformed colonies (Fig 5E).

Literature reports [19,20] and our observations of T5 exonuclease cleaving nicked circular left open the possibility that a large proportion of assembled product-length DNA is nicked circular DNA rather than the linear form. We tried to see if linear $\lambda$ DNA could be mostly circularized in the presence of Ampligase via cos site ligation, regardless of whether nicks are present due to reaction-environment induced DNA damage. The cos site of a linear $\lambda$ DNA are a pair of 12 bp 5'-overhangs that are complementary with each other [21]. Ampligase is another thermostable equivalent of Taq ligase used in Gibson Assembly [17]. Ligation of $\lambda$ DNA using Ampligase removes the seemingly irrelevant substeps in Gibson Assembly [17], such as T5 exonuclease mediated single-strand generation and polymerase based extension of annealed DNA. To see if Ampligase ligated $\lambda$ DNA is circularized at all, we purified the ligation reaction mixture with our purification approach and digested the purified reaction mixture with a combination of RecBCD, Klenow fragment, and exonuclease I. In contrast to T5 exonuclease, RecBCD does not digest nicked circular DNA (Fig 5B). While linear $\lambda$ DNA ligated at the normal concentration of 10 ng/μl was nearly-completely digested by the RecBCD cocktail, linear $\lambda$ DNA ligated at the 100 fold diluted concentration of 0.1ng/μl was resistant to digestion by the the RecBCD cocktail (Fig 5F). Presence of DNA condensation protein sso7d [22] did not significantly increase resistance to digestion by the RecBCD-cocktail (Fig 5F). Thus, multiple lines of evidence support our theory that the product-length DNA in Gibson Assembly exist almost entirely in linear form, and circularization of linear DNA is both possible and efficient at sufficiently low DNA concentrations.

While we have characterized the predominant form of product length DNA in Gibson Assembly, it remains unclear if other DNA assembly techniques also mainly produce linear DNA at the product length. Since the colony output of Golden Gate and traditional restriction cloning in our experience do not generate more colonies than Gibson Assembly, we suspect that the product-length DNA are also mostly linear for those two techniques. However, as we haven't systematically visualized the full range of DNA sequence lengths generated by those two assembly techniques, it is possible that there are substantially fewer DNA sequences having the product size and the ones having the product size are mostly circularized. A second line of evidence also supports other assembly techniques generating product-length DNA mostly in the linear form. Ampligase-mediated ligation of linear $\lambda$ DNA also resulted in mostly digestable linear DNA (Fig 5F). However, Ampligase is not used in Golden Gate or traditional restriction cloning. Thus, characterizing the amount of product-length DNA and whether the product-length DNA are predominantly linear in restriction-based cloning methods could also open up venues to improve those techniques as well.

Furthermore, circularization of linearly assembled DNA could have benefits beyond increasing colony output, such as decreasing the probability of misassembly in DNA constructs. Because double-stranded DNA ends are required to initiate homologous recombination, linear DNA ends having homology with an internal region of the assembled fragment

may trigger unintended DNA circularization to result in an *in vivo* amplifiable but misassembled plasmid. This mechanism could explain the lower percentage of correctly assembled plasmids at increased number of DNA fragments that is seen in our group. As such, it may be worthwhile to develop a method to successfully circularize linearly assembled DNA and test whether DNA circularization improves the percentage of colonies having correctly assembled DNA.

# Materials and methods

## Precipitation based purification of nucleic acids

Protein denaturation and nucleic acid precipitation is simultaneously carried out in the presence of a suitable concentration of guanidine isothiocyanate (GITC: 2–4 M prior to isopropanol addition, ≥ 4M prior to PEG addition, Fig 2E–2F) / guanide hydrochloride (GHCl: 2.5–5.3 M prior to isopropanol addition, 5.3M prior to PEG addition, Fig 2E–2F) and isopropanol/polyethylene glycol 8000 (PEG). Concentration of stock solutions are 6M for GITC, 8M for GHCl, 40% w/v for PEG, 100% for isopropanol. Different samples require different combinations of the denaturing salts and precipitation reagents, so we list the combinations for samples that we have processed below. All procedures are carried at room temperature (22 °C ) unless otherwise specified.

- For PCR purification where fragment size is greater than 400 bp, GHCl is added to the sample to 5.3M, the sample is mixed, and then 40% w/v PEG 8000 is mixed in to a final concentration of 13.3%.
- For PCR purification where fragment size is less than or equal to than 400 bp but greater than 200 bp, GITC is added to 4M and P3 (3M potassium acetate [KAc], pH4.8) is added to 7.4% (v/v), the sample is mixed, and then 40% PEG 8000 is mixed in to a final concentration of 13.3%.
- For PCR purification where fragment size is less than or equal to 200 bp, GITC is added to the sample to 2M, the sample is mixed, and then 1 volume of isopropanol is mixed in.
- For genomic DNA purification, 0.5 mL *E. coli* culture is centrifuged to remove growth media, resuspended in 118.67 μl of resuspension buffer (T50E10 + 1.25% Triton X100 + 2 mg/mL lysozyme), and incubated at 37°C for 15 min to perform crude cell lysis. 6.65 ul of 20 mg/mL of proteinase K is added to a final concentration 1 mg/mL, 1.33 μl of 10 mg/mL of RNase A is added to a final concentration of 100 μg/mL, and the sample is mixed. 6.65 μl of 6M GITC is added to a final concentration of 300 mM. The sample is mixed and incubated at 55°C for 15 min for thorough cell lysis. 267 μl of 8M GHCl is mixed-in to a final concentration of 5.3M. 0.5 volume of 40% PEG 8000 is mixed in.
- For plasmid purification, 5 mL of E coli culture is centrifuged to remove growth media and resuspended in 250 μl of resuspension buffer (T50E10 + 100 μg/mL of RNase A). 250 μl of alkaline solution (1% SDS + 200 mM NaOH) is added, and the sample is inverted for up to 5 minutes until no visible clumps remain. 350 μl of neutralization solution (4.2M GHCl + 0.9M potassium acetate, pH 4.8) is added, and the sample is inverted until the white precipitate disintegrates into pieces. The sample is then centrifuged at maximum speed to pellet the white precipitate, and the supernatant is transferred to a new tube. 1/3 volume of 40% PEG 8000 is mixed with the transferred supernatant.

If the sample contains less than 2.5 ng/μl of nucleic acids (DNA + RNA), then linear polyacrylamide (LPA, from Millipore Sigma) is added to 10 ng/μl prior to addition of the precipitation agent (isopropanol or PEG). All samples are placed in the centrifuge with a similar orientation

to keep consistent the expected locations of the usually invisible DNA pellets. Centrifugation at $\geq 14,000g$ (validated at this speed, but the actual requirement may be lower) for 20 min at room temperature is used to collect the nucleic acid pellet at the expected locations of the microcentrifugation tubes. The supernatant is then carefully withdrawn using a micropipette to minimize the chance of carrying away the easily displaceable nucleic acid pellet. If the rate of withdrawal is too fast when the pipette tip is near the DNA pellet, DNA recovery will often decrease.

To remove residual protein, at least 25 volumes of the solution containing a high concentration of chaotropic salts and isopropanol (2.67M GHCl + 50% isopropanol for $\geq 300$bp fragments; 2M GITC + 50% isopropanol for < 300bp fragments) is added to the pellet and the sample is gently inverted 8 times, to denature and dilute any residual protein. The supernatant is withdrawn using a pipette tip attached to a vacuum line after 5 min centrifugation at $\geq 14,000g$ at room temperature. Vacuum assisted aspiration is useful for maximizing the amount of supernatant withdrawn.

To remove leftover salt, a volume of 70% ethanol + MgCl2 (2.5 – 5 mM) that is at least 100 times the supernatant's residual volume is added and the sample is inverted 8 times. The supernatant is again withdrawn using a pipette tip attached to a vacuum line after centrifugation at room temperature. The sample is left at room temperature for 5 min to let residual ethanol evaporate completely.

Finally, the nucleic acid pellet is resuspended in a suitable buffer (T10E0.1) or water.

## Phenol-chloroform extraction

Please see Fig 1C and Sambrook et al [7] for an overview of the different steps for this class of purification methods. To summarize: an equal volume of phenol-chloroform is added to the sample, vortexed briefly to emulsify the organic and aqueous phase, and centrifuged at $\geq 14,000g$ for 1 min to separate the two liquid phases. The upper aqueous phase is transferred to a fresh tube. An equal volume of chloroform is added to the transferred aqueous phase, vortexed briefly to emulsify the organic and aqueous phase, and centrifuged at $\geq 14,000g$ for 1 min. The upper aqueous phase is again transferred to a fresh tube. A volume of 3M NaCl equal to the transferred aqueous phase volume is added. Another volume of isopropanol equal to the sum of the aqueous phase and NaCl volumes is mixed into the sample and centrifuged at $\geq 14,000g$ for 30 min to precipitate the DNA. The supernatant is carefully withdrawn using a pipette tip to avoid disturbing the often invisible DNA pellet. A volume of 70% ethanol + 5 mM MgCl2 that is at least 100× the supernatant's residual volume is added and the sample is inverted 8 times to dilute the residual supernatant. The sample is centrifuged for 5 min at $\geq 14,000g$ and the supernatant is aspirated away using a pipette tip attached to a vacuum line. The sample is left at room temperature for 5 min to let residual ethanol completely evaporate and the nucleic acid pellet is resuspended in either T10E0.1 or water.

## Silica column purification

Except where modifications are indicated in this manuscript, manufacturer's instructions were followed for QIAquick, QIAprep, and NEB DNA Cleanup kits. Please see Fig 1D for an overview of this class of purification methods.

## Proteinase K deactivation assay

Please see Fig 2D for an overview of the assay. Linear $\lambda$-DNA was added to 2.5 ng/µl and proteinase K (BioShop) was added to 1 mg/mL prior to addition of the precipitation agent, and

the sample was mixed. The sample volume prior to precipitation agent addition was 80 μl. The desired variant of the purification method was carried out as described above, and the resulting DNA pellet was resuspended in 10 μl of T10E0.1. 3.62 μl of the purified sample was combined with 0.05 units of Ampligase (Biosearch Technologies) and Ampligase buffer in a 5 μl reaction, and then incubated at 45°C for 1.5 h to allow residual proteinase K to destroy Ampligase. To verify that ampligase is active, the purified sample was replaced with water. 100 ng of $\lambda$-HindIII ladder (New England Biolabs) was mixed into the sample while on ice. To visualize the unmodified $\lambda$-HindIII ladder as a reference, the purified sample was replaced with water and no ampligase was added. The sample was incubated for 10 min at 45°C to let the intact Ampligase join the cos site within the $\lambda$-HindIII DNA. The sample was then immediately chilled on ice, and mixed with EDTA-containing gel-loading dye to stop Ampligase activity.

### BSA contamination assay

The presence of BSA contamination in samples purified using our method was assessed by the OD260/OD280 ratio. 4-10 mg/mL of BSA (BioShop) together with 2.5 ng/μl of 1kb+ ladder (New England Biolabs) prior to precipitation agent addition was purified using the indicated purification method. The sample volume prior to precipitation agent addition was 80 μl and the purified DNA pellet was resuspended in 5 μl of T10E0.1. For each purified sample, OD260 and OD280 were measured on a NanoQuant plate using the Tecan M1000 Pro plate reader. To verify the expected absorbance ratio of pure DNA, we measured OD260 and OD280 of the equivalent amount of 1kb+ ladder (200 ng) in 5 μl of T10E0.1.

### T5 exonuclease inhibition assay

Purified samples were incubated with T5 exonuclease (New England Biolabs) to assess inhibition of the enzyme by the purification technique. For NEB DNA Cleanup and QIAquick, 250 μl of the respective binding buffer was added directly to the column and centrifuged for 1 min. Subsequent washing steps were performed according to manufacturers' instructions. The elution volume was either 6μl or 24 μl. For our purification method, 200 ng of $\lambda$-DNA (New England Biolabs) and 80 μg of proteinase K in 80 μl prior to precipitation agent addition was purified into a 10 μl sample, where 6 μl was used in the subsequent T5 reaction. 1.5 units of T5 exonuclease and NEBuffer 4 were added to each purified sample and incubated for 30 min on ice. 250 ng of 1kb+ ladder (NEB) was then added to each T5 containing reaction on ice and incubated for 7.5 min at 37°C . To assess maximal activity of T5 exonuclease, 6 μl of the purified sample was replaced with T10E0.1.

### Exonuclease digestion assay

1.25 units of T5 exonuclease or 1.25 units of RecBCD (New England Biolabs) in a 2.5μl reaction was incubated in the presence of the appropriate buffer and the appropriate sample for 30 minutes at 37C. The sample is either 250 ng of 1kb+ ladder, 25 ng of a purified plasmid, or a purified DNA assembly reaction.

### Purification and verification of sso7d

BL21 DE3 cells containing His-tagged sso7d expression plasmid was induced and the overexpressed sso7d was purified using the Protino Ni-TED 1000 packed column. His-tagged sso7d was placed downstream of an IPTG inducible T7 promoter using NEB Hifi Assembly and

a sequentially correct colony was identified by Sanger sequencing. The purified sso7d plasmid was then transformed into BL21 DE3 cells using electroporation and the transformation mixture was plated on an antibiotic-containing plate [23]. A transformed BL21 DE3 colony was inoculated into 250 mL of antibiotic containing media and grown until OD600 of 0.6. The culture was then chilled on ice and IPTG was added to a final concentration of 1mM to induce the expression of sso7d. The culture was then grown at 22°C for 18 hours. Following expression induction, the cells were harvested using a chilled centrifuge and manufacturer's instructions on purification under native conditions were followed. Briefly, purification of His-tagged protein involved resuspension of cell pellet, sonication of resuspended cells, clarification of sonicated lysate, applying the clarified lysate to the Ni-TED column, washing the Ni-TED column, and finally eluting the His-tagged sso7d. Compared to the standard Ni-NTD columns, the Ni-TED column was chosen to maximize specificity of the column to catch only the His-tagged protein, by reducing the number of available His binding sites of each $Ni^{2+}$ ion from 2 to 1. Amicon Ultra 15mL centrifugal filter was then used to exchange the elution buffer of the purified protein sample with diluent D (10 mM Tris-HCl pH7.4, 100 mM KCl, 0.1 mM EDTA, 1 mM DTT, 0.1% Triton X-100, 50% glycerol), according to manufacturer's instructions. Protein purity was verified using SDS-PAGE (S1A Fig), according to manufacturer's instructions [24]. Protein concentration was determined using the standard Bradford assay [25].

DNA binding activity of sso7d was verified using the electrophoretic mobility shift assay [26] (S1B Fig). Specifically, 100 nM of a 70-bp oligo duplex (Eurofins Genomics) was incubated at 85°C for 2 min and slowly cooled to 4°C at -0.1°C /s to anneal the oligo duplex. Various concentrations of sso7d were added and the resulting reactions were incubated at 55°C for 10 min to allow sso7d binding to reach equilibrium. Glycerol was added to a final concentration of 5% and the resulting sample was loaded onto a 7.5% TBE native polyacrylamide gel (BioRad).

## Circularization assay

We attempted to use Ampligase to circularize linear $\lambda$-DNA (New England Biolabs) at different concentrations and in the presence versus absence of sso7d. Specifically, 50 ng of linear $\lambda$-DNA was diluted in 1× Ampligase buffer to a final concentration of 0.1, 1, and 10 ng/µl, in presence and absence of 1.78 µM of sso7d. The samples were incubated at 70°C for 5 minutes to allow sso7d to bind to the $\lambda$-DNA and suddenly chilled on ice. Ampligase was added to each sample to a final concentration of 0.02 units/µl. The reactions were incubated at 45°C for 1 hour and then chilled on ice.

We then purified all reactions using our purification method and digested the purified reactions using a mixture of RecBCD + Klenow (New England Biolabs) + Exonuclease I (New England Biolabs). Specifically, all circularization reaction volumes were first normalized to the maximum 500 µl by adding applicable amount of T10E0.1 buffer. Each sample was then purified using the GHCl + Isop variant of our purification method in the presence of 5 ng/µl of LPA and resuspended in 7.5 µl of water. To assess degree of DNA circularization, 2.5 µl of the purified sample was incubated at 37°C for 1 hour in the presence of NEBuffer 4 + 1 mM ATP, 0.375 units/µl of RecBCD, 0.027 units/µl of Klenow, 0.375 units/µl of Exonuclease I.

## Multiple comparison tests

ANOVA and Tukey's honestly significant difference test [8] were conducted in the R programming environment, using the functions 'aov' and 'glht'. Compact letter display was generated

using the function 'cld'. Two groups of data are statistically indistinguishable if and only if they share at least one letter.

## Abbreviations and definitions

| | |
|---|---|
| cfu | Colony forming units |
| CLD | Compact letter display [8] (see Methods) |
| EDTA | Ethylenediaminetetraacetic acid |
| GITC | Guanidine Isothiocyanate |
| GHCl | Guanidine hydrochloride |
| Isop | Isopropanol |
| KAc | Potassium acetate |
| LD | 1kb+ DNA ladder (New England Biolabs) |
| LPA | Linear polyacrylamide |
| NEB | NEB DNA Cleanup kit |
| PCR | Polymerase Chain Reaction |
| PEG | Polyethylene glycol 8000 |
| PEG-X | PEG 8000 at X% final concentration. |
| Precipitation agent: | Isop or PEG |
| pUC19 | Addgene #50005 is a carbenicillin resistant plasmid commonly used to assess transformation efficiency of *E. coli* cells. |
| QQ | QIAquick PCR purification kit |
| QM | QIAprep Miniprep |
| Replicates | Each replicate is an independently processed sample from the same starting materials. |
| Tris-HCl | Tris(hydroxymethyl)aminomethane hydrochloride |
| T10E0.1 | 10 mM Tris-HCl, 0.1 mM EDTA, pH8 |
| T50E10 | 50 mM Tris-HCl, 10 mM EDTA, pH8 |

## Supporting information

**S1 Fig. Purification and verification of sso7d.** (A) SDS-PAGE analysis at various stages of the protein purification process. CR means crude lysate after sonication; CL means clarified lysate prior to loading to the Ni-TED column; W means the flowthrough fraction after the wash step; F means the final eluted protein sample; FB means the final protein sample after buffer exchange. (B) Electrophoretic mobility shift assay to detect sso7d binding of oligo duplex.
(TIF)

**S2 Fig. Visualized comparisons.** (A) Visualization of Fig 2A. (B) Visualization of Fig 2C. (C) Visualization of Fig 2F. (D) Visualization of Fig 3E. (E) Visualization of Fig 3F. (F) Visualization of Fig 3G.
(TIF)

## Author contributions

**Conceptualization:** Zhe F. Tang, David R. McMillen.

**Data curation:** Zhe F. Tang.

**Formal analysis:** Zhe F. Tang, David R. McMillen.

**Funding acquisition:** David R. McMillen.

**Investigation:** Zhe F. Tang.

**Methodology:** Zhe F. Tang.

**Project administration:** David R. McMillen.

**Resources:** Zhe F. Tang.

**Software:** Zhe F. Tang.

**Supervision:** David R. McMillen.

**Validation:** Zhe F. Tang.

**Visualization:** Zhe F. Tang.

**Writing – original draft:** Zhe F. Tang.

**Writing – review & editing:** Zhe F. Tang, David R. McMillen.

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
