## [Decision Letter · Decision Letter 0]

25 Oct 2024

PONE-D-24-37731An inhibitor-free, versatile, fast, and cheap precipitation-based DNA purification methodPLOS ONE

Dear Dr. McMillen,

Thank you for submitting your manuscript to PLOS ONE. After careful consideration, we feel that it has merit but does not fully meet PLOS ONE’s publication criteria as it currently stands. Therefore, we invite you to submit a revised version of the manuscript that addresses the points raised during the review process (see reviewer comments for details).

We look forward to receiving your revised manuscript.

Kind regards,

Dharam Singh

Academic Editor

PLOS ONE

**Journal Requirements:**

Natural Sciences and Engineering Research Council (NSERC) of Canada, grant number RGPIN-2017-06795

3. Please note that your Data Availability Statement is currently missing the repository name. If your manuscript is accepted for publication, you will be asked to provide these details on a very short timeline. We therefore suggest that you provide this information now, though we will not hold up the peer review process if you are unable.

Reviewers' comments:

Reviewer's Responses to Questions

**Comments to the Author**

1. Is the manuscript technically sound, and do the data support the conclusions?

Reviewer #1: No

Reviewer #2: Yes

2. Has the statistical analysis been performed appropriately and rigorously? 

Reviewer #1: No

Reviewer #2: I Don't Know

3. Have the authors made all data underlying the findings in their manuscript fully available?

Reviewer #1: Yes

Reviewer #2: Yes

4. Is the manuscript presented in an intelligible fashion and written in standard English?

Reviewer #1: Yes

Reviewer #2: Yes

5. Review Comments to the Author

**Reviewer #1:** The authors of paper entitled “An inhibitor-free, versatile, fast, and cheap precipitation-based DNA purification method” by Tang, Z. et al. propose a method using chaotropic salts and alcohol/polyethylene glycol for simultaneous protein deactivation and DNA precipitation. This method eliminates the need for multiple steps found in traditional techniques such as phenol-chloroform extraction and silica column absorption. However, here are a few potential areas of improvement as lack of images clarity, missing quantitative comparisons, incomplete data representation etc. Authors must looks into it rigorously before re-submitting.

**Reviewer #2:** Review: Overall, the manuscript effectively presents a novel method, significant advancement in DNA purification methods, addressing key limitations of existing techniques and offering substantial benefits in terms of cost, speed, and versatility.

Minor comments/suggestions-

1. The term "protein deactivation" is used frequently. It would be beneficial to standardize this terminology to "protein denaturation" or "protein removal," which are more commonly used in molecular biology contexts, to avoid confusion.

2. Make sure that there is a consistent formatting of chemical names and units. (e.g. PEG8000 is written PEG 8000 in some places.

3. Line 260-261: "It is not definitive whether other DNA assembly techniques also generate mostly-linear DNA product-length DNA" – consider rephrasing to improve clarity, e.g., "It remains unclear if other DNA assembly techniques also predominantly produce linear product-length DNA."

4. Line 263: "do not noticeably generate more colonies than Gibson Assembly" – quantify “noticeably” for better scientific precision, if possible.

5. Line 265-267: "many fewer DNA fragments having the product size" – consider rephrasing for smoother flow: "significantly fewer product-sized DNA fragments."

6. Line 272-273: "Circularization of linearly assembled DNA could have benefits beyond increasing colony output" – consider specifying potential benefits, even briefly, to strengthen the argument.

7. Line 279-280: "we think it would be worthwhile to develop a method to successfully circularize linearly assembled DNA" – changing “we think” to "it may be valuable" would make this statement more assertive.

8. Line 288: "Presence of a suitable concentration" – perhaps clarify what is considered a "suitable" concentration. Would providing a general range help?

9. Line 301: The description of E. coli cell lysis and genomic DNA purification is detailed, but stating the specific incubation temperature range (if variable) could be useful.

10. Line 319-321: The sentence about nucleic acid pellet invisibility can be made clearer by stating the importance of tube orientation before centrifugation explicitly.

11. Line 325: "The supernatant is carefully withdrawn" – perhaps add a note about using low-binding tips to minimize sample loss, especially with invisible pellets.

12. Line 354: "At least 100× the supernatant’s residual volume" – reword for clarity, e.g., "A volume of 100 times the residual supernatant is added.

13. Terminology (Lines 260-267): The terms "product-length DNA" and "product-sized fragments" are used interchangeably. Stick to one term consistently to avoid confusion.

14. Other Assembly Techniques (Lines 261-268): You mention other techniques like Golden Gate, but there’s no data shown. It would help to include some comparison or at least mention ongoing efforts to assess them.

15. Assay Validation (Lines 377-395): It’s unclear if controls were used in the OD ratio and exonuclease assays. Mentioning controls or validation steps would make your results more convincing

6. PLOS authors have the option to publish the peer review history of their article (what does this mean?). If published, this will include your full peer review and any attached files.

Reviewer #1: No

Reviewer #2: **Yes: **Dr. Vikas Thakur

---

## [Author Response · Author response to Decision Letter 1]

20 Nov 2024

Our thanks to the reviewers for their time and thoughtful suggestions for strengthening the manuscript. Please see our point by point responses below, with our responses marked with asterisks (*). (Please also see the file Tang-McMillen-response-to-reviewers.pdf.)

Reviewer #1

The authors of paper entitled “An inhibitor-free, versatile, fast, and cheap precipitation-based DNA purification method” by Tang, Z. et al. propose a method using chaotropic salts and alcohol/polyethylene glycol for simultaneous protein deactivation and DNA precipitation. This method eliminates the need for multiple steps found in traditional techniques such as phenol-chloroform extraction and silica column absorption. However, here are a few potential areas of improvement as below:

1. Lack of Image Clarity:

Figures and Diagrams: Since you mentioned that the images are unclear, it’s important that the authors provide higher resolution or more detailed images. Figures such as the DNA electrophoresis results (gel images), and other experimental data are central to demonstrating the efficacy of the new method.

Recommendation: Authors should ensure that all figures are sharp and annotated clearly to highlight the key results. Increasing the image resolution or simplifying complex diagrams could improve reader comprehension.

* This appears to have been an issue with the submission system itself. Our own versions of the figures are clear, but the process of generating the PDF for review seems to have made them unacceptably blurry. Discussions with the editors have not yet resolved the problem, but we will submit a version of the PDF that includes the figures in the submitted PDF file itself, in the hope that these will pass through the system unchanged and thus offer clear versions of the figures. (This will likely result in there also being an extra set of (possibly blurred) figures appended at the end of the document, but the system insists on a separate upload of the individual figures, so there does not seem to be any way to proceed with the resubmission without this.)

2. Missing Quantitative Comparisons:

While the paper does mention that the new method is cheaper and faster, there are limited quantitative comparisons of yields, purity, or transformation efficiencies in the images, which could better illustrate the claimed advantages over other methods.

Recommendation: Provide side-by-side comparisons in the figures with additional data showing exact numbers (e.g., purity levels, transformation rates, yield efficiency), possibly through clearer charts or tables.

* We added a number of pieces of yield information that were missing in the previous version of the manuscript:

• Figs 3B and 3C: To provide more direct side-by-side comparisons of DNA purity, we now further compare Sanger sequencing read-length between our method and QiaPrep for the single-copy plasmid pBeloBac11. Previously, we had only pointed out the low sequence read-length of QiaPrep'ed pBeloBac11, without noting the performance of our method. There are now two distinct assays that compare DNA purity between silica columns and our method: DNA sequencing (Fig 3B) and T5 exonuclease digestion (Fig 3C).

• Fig. 3E: We added the PEG precipitation recovery percentage

• Figs 3F and 3G: We now compare single copy plasmid extraction yield between QiaPrep and our method for two different plasmids

3. Incomplete Data Representation:

The article mentions certain experiments (like comparing the inhibition of T5 exonuclease) but doesn't fully visualize them clearly. The images used to represent these findings need to communicate the results effectively.

Recommendation: Consider adding more graphical elements like bar charts or scatter plots to visually quantify data, making it easier for the reader to assess improvements over existing techniques.

5. Lack of Visualized Statistical Data:

Even though the text describes some experiments as yielding "sufficient DNA" or "better results," the visual data to support these claims in the form of histograms or other statistical visualizations are minimal or unclear.

Recommendation: Include more statistical graphs (with error bars, significance markers) to strengthen the conclusions about the efficiency and reliability of the novel method.

* We have added both compact letter display and bar plots to support our claims that one group of data has a higher metric than a second group of data. The two tools were used to show the effect of overloading the silica column on Sanger sequencing read length (Fig 2A), how recovery of chaotropic salt based precipitation can be enhanced (Fig 2C), and whether our method resulted in satisfactory yield (Fig 3E, 3F, 3G).

4. Figures are Hard to Interpret:

Some figures, like gel electrophoresis results, can be difficult for non-experts to interpret without more explanation in the captions or labeling of DNA bands, controls, or markers.

Recommendation: Provide clearer legends, annotations, and labels on each gel lane and graph so the images become self-explanatory even for those not familiar with the experimental setup.

* We have added a number of annotations to help clarify the figures.

• Fig. 2E: We have indicated which \lambda -HindIII fragment is meant to ligated by ampligase.

• Fig. 3C: Since the smallest not-fully digested band migrates lower than the original ladder, we indicated where we consider the smallest not-fully digested band to appear.

• Fig. 3F: When comparing sample migration profiles between our method and QiaPrep, we now note where the expected plasmid size is and what QiaPrep's fragment size is.

• Fig. 5A: For visualization of partial products in a Gibson Assembly reaction, we now explicitly label the sizes of different partially assembled fragments that appear on the gel image.

• Fig. 5B: When comparing plasmid sample digestion profiles between T5 exonuclease and RecBCD, we now label bands that are expected to be linear, nicked circular, or completely circular.

6. Comparative Analysis Missing in Visuals:

The comparison between silica columns and the novel method is described in the text, but better comparative visuals (e.g., side-by-side photos or simplified diagrams) would help reinforce this comparison.

Recommendation: Create summary visuals that show direct comparisons between the two methods in one image (e.g., purification time, yield, cost), to make the argument more compelling.

* As suggested by the reviewer, we have added a summary table that directly compares the different purification methods. The table gives simplified yes/no assessments (with check marks or crosses) in several categories, but also provides references to the quantitative experimental data supporting each conclusion (mainly within this paper, but referring to previous publications where necessary).

Reviewer #2

1. The term "protein deactivation" is used frequently. It would be beneficial to standardize this terminology to "protein denaturation" or "protein removal," which are more commonly used in molecular biology contexts, to avoid confusion.

* We have changed many uses of protein deactivation to protein removal, consistent with standard terminology in literature. However, we reserved the use of protein deactivation to mean using chaotropic salts to decrease protein activity. Thus, protein removal now encompass tasks performed by two different steps: protein deactivation via chaotropic salts, and protein dilution via the isopropanol + GHCl wash step.

2. Make sure that there is a consistent formatting of chemical names and units. (e.g. PEG8000 is written PEG 8000 in some places.

* We now exclusively use PEG 8000 rather than PEG8000, consistent with prior literature.

3. Line 260-261: "It is not definitive whether other DNA assembly techniques also generate mostly-linear DNA product-length DNA" – consider rephrasing to improve clarity, e.g., "It remains unclear if other DNA assembly techniques also predominantly produce linear product-length DNA."

* Thank you for the suggested rephrasing to make the sentence less awkward; we have made the revision.

4. Line 263: "do not noticeably generate more colonies than Gibson Assembly" – quantify “noticeably” for better scientific precision, if possible.

* We removed the poorly defined word “noticeably”, as we agree with the reviewer that the word is vague in that context.

5. Line 265-267: "many fewer DNA fragments having the product size" – consider rephrasing for smoother flow: "significantly fewer product-sized DNA fragments."

* We changed the awkward “many fewer DNA fragments” to “substantially fewer DNA fragments”. We wanted to reserve the word “significantly” for statistical comparisons only.

6. Line 272-273: "Circularization of linearly assembled DNA could have benefits beyond increasing colony output" – consider specifying potential benefits, even briefly, to strengthen the argument.

* We now explicitly mention the potential benefit of decreasing misassembly of DNA constructs, when DNA is correctly circularized.

7. Line 279-280: "we think it would be worthwhile to develop a method to successfully circularize linearly assembled DNA" – changing “we think” to "it may be valuable" would make this statement more assertive.

* We removed “we think” and changed the phrasing to “it may be worthwhile to”.

8. Line 288: "Presence of a suitable concentration" – perhaps clarify what is considered a "suitable" concentration. Would providing a general range help?

* We now explicitly specify the suitable range of GITC and GHCl based on the results from protein deactivation and protein contamination assays.

9. Line 301: The description of E. coli cell lysis and genomic DNA purification is detailed, but stating the specific incubation temperature range (if variable) could be useful.

* We now explicitly state that all procedures are carried at room temperature (22℃) unless otherwise stated.

10. Line 319-321: The sentence about nucleic acid pellet invisibility can be made clearer by stating the importance of tube orientation before centrifugation explicitly.

* We now emphasize orientation of centrifugation tubes prior to placing the samples in the centrifuge.

11. Line 325: "The supernatant is carefully withdrawn" – perhaps add a note about using low-binding tips to minimize sample loss, especially with invisible pellets.

* We clarified the meaning of “carefully” withdrawn to mean sufficiently low rate of withdrawal when the pipette tip is near the DNA pellet. We did not use externally low binding tips in our lab and still achieved satisfactory yield.

12. Line 354: "At least 100× the supernatant’s residual volume" – reword for clarity, e.g., "A volume of 100 times the residual supernatant is added.

* Two instances of the awkward phrasing is now rephrased to “A volume of 70% ethanol + 5mM MgCl2 that is at least 100 times the supernatant's residual volume is added”.

13. Terminology (Lines 260-267): The terms "product-length DNA" and "product-sized fragments" are used interchangeably. Stick to one term consistently to avoid confusion.

* We removed reference to “DNA fragments having the product size” and exclusively use “product-length DNA”.

14. Other Assembly Techniques (Lines 261-268): You mention other techniques like Golden Gate, but there’s no data shown. It would help to include some comparison or at least mention ongoing efforts to assess them.

* We now explicitly state as a future step the characterization of product length DNA in restriction based cloning methods.

15. Assay Validation (Lines 377-395): It’s unclear if controls were used in the OD ratio and exonuclease assays. Mentioning controls or validation steps would make your results more convincing.

* We now state in the Methods section what modifications were made to generate the appropriate controls in the relevant figures of the Results section. Explanations of controls were added for the proteinase K deactivation assay, the BSA contamination assay, and T5 exonuclease inhibition assay.

---

## [Decision Letter · Decision Letter 1]

3 Jan 2025

An inhibitor-free, versatile, fast, and cheap precipitation-based DNA purification method

PONE-D-24-37731R1

Dear Dr. McMillen,

We’re pleased to inform you that your manuscript has been judged scientifically suitable for publication and will be formally accepted for publication once it meets all outstanding technical requirements.

Kind regards,

Dharam Singh

Academic Editor

PLOS ONE

Additional Editor Comments (optional):

The revised manuscript has addressed the concerns and suggestions of two independent reviewers. Hence, teh manuscript is acceptable now. Congratulations to the authors.

Reviewers' comments:

Reviewer's Responses to Questions

**Comments to the Author**

1. If the authors have adequately addressed your comments raised in a previous round of review and you feel that this manuscript is now acceptable for publication, you may indicate that here to bypass the “Comments to the Author” section, enter your conflict of interest statement in the “Confidential to Editor” section, and submit your "Accept" recommendation.

Reviewer #1: All comments have been addressed

Reviewer #2: All comments have been addressed

2. Is the manuscript technically sound, and do the data support the conclusions?

Reviewer #1: Yes

Reviewer #2: Yes

3. Has the statistical analysis been performed appropriately and rigorously? 

Reviewer #1: Yes

Reviewer #2: I Don't Know

4. Have the authors made all data underlying the findings in their manuscript fully available?

Reviewer #1: Yes

Reviewer #2: Yes

5. Is the manuscript presented in an intelligible fashion and written in standard English?

Reviewer #1: Yes

Reviewer #2: Yes

6. Review Comments to the Author

Reviewer #1: Thank you for thoroughly addressing the feedback provided in the previous review. The inclusion of the summary table that directly compares the different purification methods is a significant improvement to the manuscript. By using simplified yes/no assessments with check marks or crosses, the table effectively conveys key distinctions among the methods, making the comparisons intuitive and easy to follow.

Moreover, the detailed referencing of quantitative experimental data to support each assessment adds depth and scientific rigor to the presentation. Referring not only to the data within this paper but also to relevant prior publications provides a well-rounded and comprehensive basis for the conclusions drawn. This approach enhances the credibility and utility of the table, allowing readers to better understand the rationale behind the comparisons.

Overall, this revision has greatly improved the clarity, interpretability, and accessibility of the results, addressing the concerns raised in the earlier review. The manuscript is now better positioned to contribute meaningfully to the field.

Reviewer #2: (No Response)

7. PLOS authors have the option to publish the peer review history of their article (what does this mean?). If published, this will include your full peer review and any attached files.

Reviewer #1: **Yes: **Dr. Mohit Kumar Swarnkar

Reviewer #2: **Yes: **Dr. Vikas Thakur

---

## [Editor Report · Acceptance letter]

PONE-D-24-37731R1

PLOS ONE

Dear Dr. McMillen,

I'm pleased to inform you that your manuscript has been deemed suitable for publication in PLOS ONE. Congratulations! Your manuscript is now being handed over to our production team.

Kind regards,

on behalf of

Dr. Dharam Singh

Academic Editor

PLOS ONE